# What Sets Physically Active Rural Communities Apart from Less Active Ones? A Comparative Case Study of Three US Counties

**DOI:** 10.3390/ijerph182010574

**Published:** 2021-10-09

**Authors:** Christiaan G. Abildso, Cynthia K. Perry, Lauren Jacobs, M. Renée Umstattd Meyer, Megan McClendon, Michael B. Edwards, James N. Roemmich, Zachary Ramsey, Margaret Stout

**Affiliations:** 1Department of Social and Behavioral Sciences, School of Public Health, West Virginia University, Morgantown, WV 26506, USA; zramsey@mix.wvu.edu; 2School of Nursing, Oregon Health & Science University, Portland, OR 97239, USA; perryci@ohsu.edu; 3School of Kinesiology and Physical Education, College of Education and Human Development, University of Maine, Orono, ME 04469, USA; lauren.jacobs@maine.edu; 4Department of Public Health, Robbins College of Health and Human Sciences, Baylor University, Waco, TX 76798, USA; Renee_Umstattd@baylor.edu (M.R.U.M.); megan.mcclendon@ag.tamu.edu (M.M.); 5Department of Parks, Recreation and Tourism Management, College of Natural Resources, North Carolina State University, Raleigh, NC 27695, USA; mbedwards@ncsu.edu; 6US Department of Agriculture, Agricultural Research Service, Grand Forks Human Nutrition Research Center, Grand Forks, ND 58201, USA; james.roemmich@usda.gov; 7Department of Public Administration, West Virginia University, Morgantown, WV 26506, USA; margaret.stout@mail.wvu.edu

**Keywords:** rural health, physical activity, positive deviance, qualitative research, community capital framework, comparative case study

## Abstract

Background: Rural US communities experience health disparities, including a lower prevalence of physical activity (PA). However, “Positive Deviants”—rural communities with greater PA than their peers—exist. The purpose of this study was to identify the factors that help create physically active rural US communities. Methods: Stakeholder interviews, on-site intercept interviews, and in-person observations were used to form a comparative case study of two rural counties with high PA prevalence (HPAs) and one with low PA prevalence (LPA) from a southern US state, selected based on rurality and adult PA prevalence. Interview transcripts were inductively coded by three readers, resulting in a thematic structure that aligned with a Community Capital Framework, which was then used for deductive coding and analysis. Results: Fifteen stakeholder interviews, nine intercept interviews, and on-site observations were conducted. Human and Organizational Capital differed between the HPAs and LPA, manifesting as Social, Built, Financial, and Political Capital differences and a possible “spiraling-up” or cyclical effect through increasing PA and health (Human Capital), highlighting a potential causal model for future study. Conclusions: Multi-organizational PA coalitions may hold promise for rural PA by directly influencing Human and Organizational Capital in the short term and the other forms of capital in the long term.

## 1. Introduction

Physical activity (PA) is widely recognized as an important determinant of both physical and mental health [1] and a critical public health target [2]. Identification of the population-level determinants of PA has received increasing attention over the past 20 years and is often framed using social ecological theory [3]. Focusing on the influence that policy and built environment changes have on PA [4], the evidence displays a decidedly urban bias [5,6,7]. This bias in evidence-based community-level interventions that increase access to safe places for PA [8] is based on the supposition that there exist human, political, and financial resources in a community to make changes to planning documents, local policies, and investments in development. This assumption of existing resources cannot safely be made in rural areas of the US that continue to see a widening income gap when compared with urban areas and have only recently reversed decades-long population declines [9]. The body of literature also neglects the strength and overlapping nature of social networks in rural communities, which may help overcome the limited human, political, and financial resources to allow communities to move much more quickly and with greater influence than in urban areas [10]. In summary, the influential urban-centric, social-ecological models that dominate our understanding of the environmental determinants of PA may not have the same applicability in rural areas and small towns.

Identifying ways to increase PA and improve quality of life in rural areas is critical. Rural communities in the United States (US) experience health inequities and disparities [11], including shorter life expectancy [12], greater mortality rates [13], greater prevalence of obesity [14,15], higher all-cancer age-adjusted death rates [16], and lower rates of engaging in critical preventive health behaviors such as PA [17,18,19].

Despite being identified as a gap in previous literature reviews, effective rural-specific, community-level PA policy, systems, and built environment change strategies are still yet to be clearly identified [5,6,7]. However, some approaches recommended for urban and suburban areas have been adapted for rural areas [20], and barriers to change in policy and built environment in rural areas have been identified [21]. Rural-specific evidence is uncommon in the research because of the urban-centric lens that is focused almost exclusively on political and built environment change without considering the systems-level structural capacity constraints in rural areas, such as financial, organizational, and human resources [22].

Though adults in rural counties on average have lower rates of PA than urban ones, some rural counties achieve a greater prevalence of PA than others [23]. These positive outliers serve as fertile ground for discovering rural-specific evidence and practice-based strategies. This “positive deviance” approach involves finding “uncommon, beneficial practices” used by positive outliers in resource-poor communities that help those communities achieve better results than “similarly impoverished neighbors” [24]. Such practices are likely more “affordable, acceptable, and sustainable” by peers because they are used by comparable communities [24]. Commonly applied outside of the US, this approach has been used to identify positive nutritional practices to combat malnutrition in rural Haiti [25] and rural Vietnam [26] and among Afghan refugees in Pakistan [27]. Similarly, the “Blue Zones” project has identified places around the world with the greatest rates of people living to age 100 years and the behaviors that these people employ [28,29]. Logically, rural US counties that stand out as positive outliers in terms of the prevalence of PA can be studied to identify the beneficial practices unique to active living in rural areas. Thus, the purpose of this study was to identify the factors that influence PA in those rural communities that achieve greater-than-expected rates of PA. Achieving this purpose may inform the development of a rural-specific model and the identification of best practices in rural PA promotion for subsequent dissemination and implementation.

## 2. Materials and Methods

We conducted a comparative case study of two positive outlier counties and one negative outlier county in a southern US state. Each county served as a “case” (i.e., a bounded system) that was studied in depth [30,31]. Inclusion criteria were based on county-level rurality and prevalence of PA. The rurality inclusion criterion was set as the counties in this state with the highest decile of population living in a rural area according to the 2010 US Census. Counties that met the rurality inclusion criterion were then classified as a positive outlier or negative outlier based on county-level PA guideline prevalence for adult men and women from the 2011 Behavioral Risk Factor Surveillance System (BRFSS) using Dwyer-Lindgren and colleagues’ estimates [23]. Guidelines at that time were 150 min of moderate PA per week or 75 min of vigorous PA, or a combination of the two (i.e., 1 min of vigorous = 2 min of moderate). Each county in the top quintile was classified as a positive outlier, hereafter a “high physical activity” (HPA) county; each county in the bottom quintile was classified as a negative outlier, hereafter a “low physical activity” (LPA) county. This list was further narrowed to remove those with only one incorporated area (e.g., city, town), effectively eliminating counties with <2000 residents. In collaboration with a statewide partner, two HPA counties (HPAs) and one LPA county were identified that represented the sociodemographic, economic, and cultural diversity of the rural counties in the state. This permitted a comparison between two very distinct HPAs that represent the diversity of rural places, in addition to the comparison between HPAs and the LPA.

The research team collected data in the county seat and the incorporated area with the largest population that was not the county seat. The population of the county seats ranged from 411 to 2480, and the other incorporated areas from 223 to 2500. Characteristics of case study counties are displayed in Table 1, highlighting the diversity of race/ethnicity, income, and economy, despite 100% of the population in each county residing in a rural area. HPA1 and HPA2 were the most and fourth-most active of the eligible counties in the state, respectively; the LPA was the second-least active. The prevalence of achieving PA guidelines was at least 20% greater in the HPAs than the LPA.

### 2.1. Participants

The County Coordinator for the Cooperative Extension office in each county was identified as the key stakeholder, because they and the other Extension Agents are generally very knowledgeable about influential people, places, and activities in their county. Contact was initiated with each of the three County Coordinators by an email from a state-level Extension Specialist. The research team followed up within a week, at which time the study was described and voluntary approval to participate was solicited. All three County Coordinators agreed to participate. The County Coordinators and Extension Agents were interviewed and involved with data collection in their county. Snowball sampling was used to identify other key stakeholder interviewees by asking the County Coordinator to identify and contact key people involved in encouraging local residents to be physically active. In addition, the research team asked each key stakeholder interviewed to identify additional people to interview during the data collection visits.

Fifteen key stakeholder interviews and nine intercept interviews were completed, and on-site observations were conducted in all six municipalities. Details of each stakeholder interview participant are presented in Table 2, including an Interviewee Code that was assigned to each based on location and interview number. Results of the analysis of key stakeholder interviews are presented with illustrative quotes and/or triangulated support from observations and/or intercept interviews noted. Direct quotes are assigned to participants using the Interviewee Codes described in Table 2.

### 2.2. Instruments and Data Collection

Data were collected via digitally recorded semi-structured key stakeholder interviews, intercept interviews, and in-person observations during a nine-day period in November 2017. Two research team members collected data during two- to three-day visits to each of the three counties. Stakeholder interviews were conducted in a private location convenient for each stakeholder, usually at their workplace, after an introduction by the County Coordinator. A semi-structured interview script of four questions was used to identify and probe the facilitators and barriers to PA, including (1) the key people “that get people to be physically active or do exercise”, (2) the “types of activity, exercise, or sports that are popular”, (3) the locations where people are active, and (4) the barriers that “keep people from being more active”. Stakeholder interviews were professionally transcribed. The research team leader verified the transcripts against recordings and removed identifying information prior to analysis. These stakeholder interviews were the primary data source for analysis.

Intercept interviews and in-person observations were used to triangulate the stakeholder interview data. Intercept interviews were conducted with community members in community settings, such as restaurants, hotels, sports fields, and other informal settings that the research team visited while making observations, without the presence of the County Coordinator. Potential participants were approached, and those who agreed were asked about the same four topic areas from the stakeholder interview but in a very brief 3- to 5-min format, without probing for details. An additional item about PA programs and cost was included. Notes were taken during the interview. In-person observations included a driving tour of the key PA locations (e.g., parks, fitness facilities, and schools) as determined by the County Coordinator as well as walking observations of the quality of physical infrastructure conducted by the research team, independent of local stakeholders. Field notes were completed to summarize the intercept interviews and observations. The study was approved for the protection of human subjects by the West Virginia University institutional review board (protocol #1711836161).

#### Researcher Reflexivity

The research team included members from multiple states to provide a check on the bias of the analysis, which was conducted by three researchers from outside the state where the data were collected. Additionally, data collection was done in pairs that included a local researcher and a researcher who had never been to the area where the data were collected. This balance in the research team was established to prevent any single researcher’s experience or bias from affecting the data collection or interpretation of the data.

### 2.3. Analysis

Analyses were conducted to identify the factors associated with PA in the HPAs in comparison to the LPA. Our purpose was exploratory, rather than confirmatory of an existing model or theory. Thus, we used a two-step process to, first, use inductive coding to develop a codebook and, second, apply that codebook to the interview transcripts using deductive coding.

#### 2.3.1. Codebook Development

Our first step in developing a codebook used an inductive approach to allow the emergence of a code structure that was grounded in the participants’ experience [32,33]. Three researchers (CGA, LJ, and CKP) were each assigned all transcripts from one county for inductive coding. The codes were then discussed by the research team until reaching consensus on the final coding structure and operational definitions. The structure that emerged from inductive coding closely aligned with the forms of capital defined in the Community Capital Frameworks (CCFs). Therefore, we operationalized CCF definitions using the results of inductive codes oriented specifically toward PA.

#### 2.3.2. The Community Capital Framework

Various Community Capital Frameworks (CCFs) include a well-rounded set of factors to understand community phenomena [34,35,36]. As conceptualized by various community development scholars, forms of capital include cultural, human, social, organizational, political, financial, natural, and built assets or gaps [10,35,37,38]. Depending on the focus, these factors are grouped into categories understood as (1) the social, economic, and environmental determinants of health [39]; (2) the indicators of community quality of life, livability, and resilience [40,41,42,43]; (3) the elements of sustainability—people, prosperity, and place [44].

Most basically, these forms of community capital act as building blocks for one another [45]. The natural environment shapes human settlements, creating opportunities and constraints for why people live in a particular place. Culture guides how people develop and live together, including the history of the place and its people, along with their traditions, belief systems, language, foodways, creativity, and attitudes. Human capital is the foundation of a community’s capacity for development, including the residents’ health, skills, knowledge, and abilities. Social capital is the quality of relationships among individuals and groups that either fosters or hinders community capacity. Community capacity is built through organizations’ effective structures, policies, plans, and track records of fulfilling community needs and working together. Community-led development efforts require residents to have political efficacy and agency in the policymaking process. Sufficient basic income, the ability to build wealth, access to investment and lending, philanthropy, and local business reinvestment are all required financial resources for an adequate quality of life. Finally, built capital includes infrastructure, structures, and the way spaces between and around structures are designed and imbued with a sense of place. These forms of capital also interact in complex patterns that can either “spiral” a community toward or away from quality of life [10]. These interactions can be conceptualized in causal models when applied to specific phenomena.

#### 2.3.3. Transcript Coding

In the second step of analysis, three researchers (CGA, CKP, and LJ) used the codebook and operational definitions developed during the initial coding process (see Table 3) to code all transcripts in a deductive (confirmatory) manner. The interrater agreement was assessed by having the three researchers independently code one transcript from each county until consensus on the final coding structure was reached. This took two rounds of coding and consensus building. The researchers coded the transcripts by having a primary reader code the transcript, a secondary reader review the primary reader’s coding, and a tertiary reader resolve any conflicts. All coding was conducted using Microsoft Word comments to highlight text and label it with a parent code (the eight forms of Capital) and any subcodes and whether that code/subcode was present or absent. For example, if an interviewee said “we would love to have a public pool, but our parks and rec department hasn’t built one”, the text would be highlighted and coded as “Built Capital/Public Pool/absence”. That statement would also be coded as “Financial Capital/absence” if the interviewee indicated that lack of funds was a barrier. Our analysis described what forms of capital were present or absent in each case so that the presence or absence could be compared between the HPAs and LPA. Each of the three researchers were assigned as primary reader of all transcripts from one county, secondary reader of all transcripts from another county, and tertiary reader for a third county.

#### 2.3.4. Member Checking

Member checking was accomplished by sharing a summary of the preliminary findings by email with the key stakeholder from each county, including a comparison of findings from that stakeholder’s county with the other two counties. This was done to obtain confirmation or suggestions about our findings and to ensure that the research team presented an unbiased account of the data. There were no edits to preliminary analyses after member checking.

## 3. Results

### 3.1. Cultural Capital

Differences in Cultural Capital were found between the HPAs and LPA regarding (1) attitudes and beliefs about the common types of PA and (2) reasons that adults and children engage in PA. Walking was a primary form of PA in all three counties, but the HPAs had more diverse types of adult PA engagement beyond walking. For example, road biking, running, road races (running or bicycling), water sports, swimming, and golf were all common activities in HPAs in addition to walking. It is worth noting that these outdoor activities are dependent on Natural or Built Capital (e.g., pools, golf courses, lakes, and safe roads).

The reasons people engaged in PA differed as well. HPA interviewees, particularly men, highlighted an attitude that occupational/purposeful activity was influential for them, whereas “working out” was more common in the LPA. During intercept interviews in HPA2, three separate men noted the influence of occupational/purposeful PA. One stated, “I work behind a computer all day at the body shop, then come home to work” (tending show animals and renovating an old barn and farm house). An EMT described, “I work all day then come home to work on my farm; Sunday is leisure day with football”. A county official succinctly stated, “it’s gotta be work, no leisure”. In HPA1, a key stakeholder introduced the county as one that was active for “pickin’ grapes and haulin’ hay”. If engaging in leisure PA—usually walking—the primary purpose was to spend social time with family and friends. In HPA1, leisure PA was also achieved through social/group activities such as golfing, softball, and watersports. In HPA2, golf was also mentioned as a common leisure activity. HPA2.3, a retired male, also highlighted a shift toward embracing leisure PA with age:


*I used to get a lot of exercise just from some of the work I did, construction, that thing. Probably, being in the Army and when I was a kid … I did a lot of stuff as a kid, did a lot of hunting and walking. And just probably built my body up enough to last a while. I don′t know how much longer it′s gonna last … But I had a feeling because of that exercise I got as a young guy and some of the employment that I chose, self-employment, that I just stayed pretty well healthy. Now I sit around too much, but that′s why I try and go play golf two or three times a week. And go to the football and basketball game.*


While our purpose in the interviews was to discuss adult PA, interviewees frequently focused on children’s PA, because, as the previous quote from HPA2.3 highlights, many of the interviewees were raised to be active and believed that the PA skills and habits learned in youth transferred to adulthood. For children, school programming, school sports, and competitive sports leagues outside of school were the primary reasons for engaging in PA in all three counties. What differed was the additional recognition of children being active for unstructured play and transportation purposes in the HPAs, which was supported by the in-person observations. As HPA2.2 (a high school principal) stated,


*I would say, with our little kids, they′re pretty active. The summer time and in the evenings, you′re gonna see kids on bikes, kids walking, kids hanging out playing, like back in the old days, whenever kids would go outside and play. There′s quite a bit of that. I mean, now of course, as you get older, our high school kids, not so much … our high schools kids are mostly involved in sports through the school … at least 50%, probably more between 60 and 70 are involved in athletics of some sort.*


Unstructured play was recognized by HPA1.4 as well: “See, we have the parents working in (large city 50–60 miles from the county seat in adjacent county) or whatever, and so the kids play on the courts until the parents get off work”. In addition, active transportation among youth was common in the HPAs. A participant in HPA1 described a daily road closure at an elementary school in one municipality for children walking and biking to school, and an HPA2 participant (school principal) described that it was common for children to walk or bike to the elementary school. These were triangulated through in-person observation.

It is important to note that Cultural Capital—the beliefs about appropriate/common types of PA—was subject to influence by in/outflows of Human and Organizational Capital in all three counties. In the HPAs, traditional beliefs about PA established by influential people and organizations with a long track record of PA promotion were being strained by recent changes (i.e., school district leadership turnover, loss of major employer). HPA1.1 and HPA2.1 (each with over 15 years of experience in their current job with Cooperative Extension) both noted they “no longer see obese kids” thanks to long-running programming conducted in the schools. Conversely, in the LPA, personnel changes and additions to key organizations (e.g., county elected officials, Cooperative Extension) in the previous 5–7 years had an influence in shifting toward a culture of being active rather than just addressing poverty and basic needs, which was the dominant focus previously.

### 3.2. Human Capital

Most fundamentally, PA is an indicator of Human Capital, given its evidence-based status as a determinant of health at either the individual or community level of analysis [1]. However, this study focused on the individual and collective capacity to lead the community in being physically active. The analyses revealed (1) commonalities in who the influential community leaders were, (2) differences in the level of and constraints on these leaders’ involvement in encouraging PA, and (3) differences in the supports and constraints on the supply of human capital that could be engaged in encouraging PA.

Common in all three counties was the identification of school administration, school personnel (especially coaches), Cooperative Extension employees, church leaders, medical professionals, elected officials, and non-profit leaders (e.g., Boys and Girls Club Director) as influential community leaders. The key variation between the HPAs and the LPA was the degree to which these leaders focused their efforts on promoting PA. In the HPAs, church and medical leaders were involved in providing facilities for PA (Built Capital) and programming (Organizational Capital). In the LPA, those leaders were not focused on PA.

Unique to the LPA were limitations to the amount of involvement of multiple PA leaders (a) because they lived outside the county, (b) because of a county-level focus on substance use disorder and social service needs, or (c) because of a lack inclusion of Black residents in leadership in the LPA. In the LPA, Black residents make up about 50% of the population of the county seat, yet the city and county leaders were almost exclusively White. This may have manifested as a lack of facilities and/or programming desired by Black residents, except for a Boys and Girls Club with a 20-year history but no facilities of its own, whose director/founder is Black and teaches in the local high school where the programming is conducted. Also unique to the LPA was the heavy influence of an elected official (county judge) and a non-profit leader (Boys and Girls Club Director), possibly due to the long-running focus on social services and persistent poverty. As LPA.2 succinctly stated, “physical activity is a basic need”, and she has shifted her personal and organizational focus to include the promotion of PA and health to prevent the social ills associated with substance use disorder. She was very influential in helping to secure grant funding (Financial Capital) for new facilities (Built Capital) and staff (Human Capital) to increase the capacity to deliver PA programming in recent years due to her belief that PA and health are basic needs.

A critical capacity constraint in many rural counties in the US is the limited supply of human capital (i.e., number of people available) to lead PA programming and the resulting oversized influence—both positive and negative—that each person can have. This theme was witnessed in a statement by HPA1.3 when discussing how recent school administration changes (installing new leaders for whom PA was not a high priority) resulted in reductions to physical education and health teaching staff:


*I just think that we [school system] lost some key funding, and they went to make some cuts. The people above my head are not as strong of a fitness and health people as I used to have here, including we got a new superintendent, we got a new curriculum director, we got three new principals.…I already broke them other ones in too…I don′t want to say I bullied them into it, but I persuaded them in a positive way to say, “Look, this is the best thing for all humans. If humans were meant to sit down, we′d be built a different way.” They′re not.*


Time constraints may also have been a critical barrier to increasing the quantity of human capital focused on PA. However, this time constraint was as result of differing reasons when comparing the HPAs to the LPA. In HPA counties, a lack of time to engage as a leader in promoting PA was due to long commute times to nearby metropolitan areas. In the LPA, the lack of time was due to the inconsistency of available time due to hourly, temporary, and/or shift work. As LPA.7 said,


*Some [shifts] are 8:00 [A.M.] to 8:00 [P.M.]…you know if they get a chance to work over, a lot of parents I know work over. They won′t know until that day starts, you know. And if they need to work over, they′ll work over… [because of that] we don′t have a lot of participation with hands on at the Boys and Girls Club with African American dads.*


This was echoed by a hotel desk worker in the LPA, a middle-aged, Asian-American female, who said during an intercept interview that she thought lack of time was the problem because many people worked evening/night shifts at one of the three nursing homes or the hospital, which were major employers. The limitations on time for PA leaders to engage in creating a physically active community and for increasing the supply of people engaged in being PA leaders also created a barrier to creating Social and Cultural Capital.

### 3.3. Social Capital

Social Capital reflects the social “glue” to make things happen that increase community PA levels. What emerged from the analyses was a theme of “us and them” barriers to creating Social Capital in all three counties, with differences between the HPAs and LPA in response to these barriers due to the differing track record and representativeness of key PA leaders. Barriers included concerns about an infusion of tourists and second homeowners in HPA1 and the loss of a large employer in HPA2, and the resulting loss of population, infusion of temporary workers for short-term work, and daily infusion of school students and teachers to fill the local school.

What differed when comparing the HPAs and the LPA was the response to “us and them” concerns. The response in the HPAs was to rely more heavily on the PA leaders with longstanding relationships and collaborations in the community (Human and Organizational Capital) to maintain the social norm that PA was expected (Cultural Capital). Interestingly, this manifested in HPA1 as a “locals only” PA location (i.e., a swimming hole), while still embracing the activities preferred by the outsiders (i.e., building bike lanes along popular bike routes). In HPA2, the response to the population fluctuation was to rely more heavily on female leaders in Cooperative Extension and the school system with a long history of collaboration.

The track record of key PA leaders (Human Capital) and organizations (Organizational Capital) to successfully support community PA appeared to have an influence on the response to these Social Capital concerns. In the HPAs, the PA leaders and organizations had long-standing track records of PA influence individually and collectively, generally 15+ years. In the LPA, a few of these key people had only 5–7 years of experience in that county. That collective, lengthy track record in the HPA manifested as important PA facilities such as parks and schools, often due to collaboration across organizations (e.g., a public-private partnership in HPA1 to access water rights to support the construction of a golf course using private capital, which was free for the high school’s golf team to use). In the LPA, the limited track record of a few key individuals was just beginning to yield positive results (e.g., securing grant funding to build new facilities), but those effects were not yet widespread throughout the county to all residents, possibly because of the lack of representation of Black residents in these collaborations.

### 3.4. Organizational Capital

Analyses revealed the need to collaborate across local organizations, particularly to help overcome the lack of staff capacity (e.g., no grant writer or program/events leaders) and financial capacity to support programming and facilities. Key differences emerged with respect to (1) the types of organizations in existence, (2) the length of collaboration of key organizations, and (3) the policies of key organizations.

One critical difference between the HPAs and LPA was the existence of a Parks and Recreation Department. The largest city in both HPAs had a Parks and Recreation Department, while the LPA had none, likely resulting in the greater quantity and quality of parks facilities (Built Capital) in the HPAs. Another difference was the involvement of a local economic development entity. In HPA1, the county and the county seat each had economic development plans that included pedestrian and bicycle facilities in the municipalities (county plan) and creating and promoting youth recreation opportunities (county seat plan). In HPA2, the most populous city had an economic development entity that raised funds to build a local health clinic (with a fitness facility and PA programming) and local parks. The LPA did not have any local economic development entity. Rather, the county was part of an 11-county regional economic development collaborative. Only in recent years had the county’s elected officials engaged with the regional economic development collaborative to help, for example, to provide matching funding for state grants to build facilities (i.e., the Cooperative Extension facility). However, the region’s economic development plan did not include a focus on recreation or PA facilities (Built Capital).

Organizational partnerships tended to have a longer duration in the HPAs, resulting in greater participation in programming in HPAs than in the LPA, where collaboration was more nascent. This was evident in repeated comments from people in the LPA about the struggle to recruit people to attend health fairs or participate in formal programming, whereas HPA1.1 clearly articulated that


*It [a worksite wellness program] went over well if it was in a group that was already established, but if I were to just market it through the newspaper, I mean, no one′s … It would have to be a group, like I′ve done it with teachers before at a school and they loved it because they were competing against other schools or the superintendent′s office or something like that, but I′ve not done a lot of physical activity education.*


The formality and level of restrictions placed on the use of school facilities such as the track, sports fields, and indoor courts for PA (i.e., shared-use policies) also differed between the HPAs and LPA. In the HPAs, no formal shared-use policies were mentioned when discussing community use of school facilities or parks. When informal “policies” were discussed, they were generally verbal requests by community members. For example, HPA1.3 said, “People are real honest and forthcoming about asking, ‘Hey, coach, is it OK if I can get a few laps in’ or whatever it may be”. A school principal in HPA2 simply stated “No, we encourage people to come use the track” when asked if they had any restrictions on the use of their facilities.

However, in the LPA, there were formal policies, with tighter restrictions on shared-use of school facilities. For example, in one of that county’s high schools, the indoor facilities were only allowed for community activities if one of the community organization’s leaders worked in the school. In the other high school in that county, formal policies were in place even for the Boys and Girls Club, whose director also worked in the school. As LPA.3 described,


*We have a board policy. If it′s a private individual the policy is much more strict, of course, because of liability reasons. You have to have a school district employee present, and there′s a deposit that must be left. There′s certain rules you have to follow when you′re here, for example, absolutely no alcoholic beverages or smoking. When it′s a community event we′re much more receptive to allowing things to happen because it′s promoting, it′s for the community, it′s for our kids. Let′s say a 30 year class reunion that wants to rent the cafeteria is a lot different than the community recreation wanting to play track and field on a Saturday for a tournament. There is a school district policy that mandates that.*


### 3.5. Political Capital

Political Capital reflects the amount of power and influence exerted by leaders to support PA as well as the access to power brokers at local, county, state, and federal levels. The strength and stability of the support of PA by leaders was greater and more stable in the HPAs than the LPA, until recent changes in school district administration in HPA1 and the loss of a major employer in HPA2. The “access to local power brokers” was readily apparent and very broad-reaching in the HPAs, as evidenced by the many different organizations that collaborated in PA. However, in the LPA, the lack of representation of Black residents in the political power structure may have prevented access to power brokers and numerous other forms of capital. Access to elected political officials was rarely mentioned in the HPAs, but in the LPA an elected official was at the core of many of the PA initiatives. However, that elected official had been in that position for seven years and had only shifted the focus of powerful community leaders away from social services toward integrating PA about five years prior to the data collection. As LPA.2 stated, there was a growing awareness among influential leaders of integrating PA to address social service needs, particularly for addressing concerns such as substance use disorder:


*I would love to have a gym kind of thing where we could bring people in and have equipment, moderate passive equipment… [for] recovering addicts. When you are in treatment, to have something like that, to be able to go and do bodybuilding, clearing your mind… When your body is healthy and your mind is active you just have a better life. And the ones around you have a better life.*


### 3.6. Financial Capital

The HPAs and LPA differed with regard to the source of the financial resources used for PA programs and infrastructure. Consistent, stable sources of local public funding were relied upon in the HPAs, whereas the LPA relied more heavily on private and foundation funding. Local tax revenue was used to run City Parks Departments, build or improve school sports facilities, and match state-level funding (e.g., Land and Water Conservation Fund grants) to support park construction in each of the HPAs (e.g., a pool and skate park in a City Park in HPA1). The LPA was more heavily reliant on private and foundation funding because of inconsistent local public funding (i.e., city and county budgets). Examples from the LPA include (1) a special levy to build new sports facilities and improve other school facilities had failed in one of the school districts in the LPA just before the research team’s visit, (2) private donations were used to help start the Boys and Girls Club over 20 years prior, and (3) regional foundations were relied upon to match Community Development Block Grant (CDBG) grants to construct sidewalks in its central business district and to build the social services center that housed the Cooperative Extension offices. This CDBG funding requires a competitive application (Human Capital), and is administered by the regional economic development collaborative (Organizational Capital), with which the local elected official became heavily involved in the seven years since being elected.

### 3.7. Natural Capital

The analyses did not reveal notable differences between the HPAs and LPA with respect to Natural Capital. Rather, all three counties were similar in the use of the outdoors for hunting and fishing for recreation, weather as a barrier to PA (i.e., rain and heat), and a concern that illegal activities (e.g., making/using illicit substances) occurred in wooded areas, as barriers to using Natural Capital for PA.

### 3.8. Built Capital

There were differences between the HPAs and LPA with respect to Built Capital, particularly regarding (1) the quantity of outdoor parks and recreation facilities and (2) the amenities at those facilities. When asked about barriers to PA, both LPA.1 and LPA.2 simply stated “facilities” and ticked off a wish list that included both outdoor (i.e., pools and parks) and indoor facilities (i.e., community center, gym, and senior center). Outdoor facilities that existed in each of the HPAs but not the LPA included pools, golf courses, walking trails, and city and state parks with playgrounds, pavilions, sports fields, and courts open to the public. The outdoor facilities in the HPAs also included amenities to overcome natural barriers to PA, such as shade from the sun at the pools and pavilions, shelter from rain (e.g., outdoor church basketball courts under a permanent pavilion), and/or lit outdoor facilities for PA during cooler evening hours (e.g., high school track with motion-activated lights and covered church courts with lights). In the HPAs, at least one of the high schools had an “extra” facility (i.e., track, field, or gym) with greater community accessibility, which was attained after a new facility was built for school teams. In contrast, in the LPA, one of the two high schools built new indoor and outdoor facilities in last 10 years but did not retain the old facilities. The other high school did not have a football field or track.

The HPAs had multiple, well-maintained public parks and trails, primarily in the largest municipality, many of which were walking distance from the downtown. Specifically, HPA1 had a city-operated park between the downtown and high school in its most populous municipality, which had an event center, swimming pool, skate park, playground, walking/jogging trail, tennis courts, baseball and softball fields, and a pavilion. It also had two state parks along a river, inclusive of swimming and flatwater paddling/boating access, hiking, biking, and horseback riding trails, and a national historic site with a walking trail. HPA2 had city-operated parks in its most populous municipality, including greenspace on its town square, and a swimming pool, playground, two walking paths, two little league fields, and a nine-hole golf course, all within two-thirds of a mile of the town square. The LPA had no public parks, trails, or recreation facilities in any of its municipalities. It did have two large national forests and reservoirs, but these had fee-for-use camping and marina facilities, with the closest swimming beach/playground facility a 25-mile drive from the most populous municipality’s downtown. When comparing the HPAs with the LPA, it was apparent that the LPA lacked Built Capital for PA. As LPA.2 stated,


*We′re building a park back behind the [Extension office] facility, and that′s a start. If we had a real senior citizen′s center that could be used where we would have indoor pool which we could do water therapy. I would love to have a gym kind of thing where we could bring people in and have equipment, moderate passive equipment…*


## 4. Discussion

The purpose of this exploratory study was to identify the factors that influence PA in rural communities that are positive outliers in achieving greater-than-expected rates of PA. Thus, the differences that emerged between the HPAs and the LPA are critical. Relative to the LPA, the HPAs had (1) a culture/social norm that adults should engage in a greater variety of outdoor, lifetime types of PA; (2) a greater quantity and quality of Built Capital in place to support the multitude of activities; and (3) a greater quantity of Human and Organizational Capital, with a longer track record of focus on PA. These findings are discussed below, followed by discussion of a potential pathway/theoretical model that emerged, suggesting that Cultural and Human Capital are crucial elements and may generate differences in Social, Built, Political, and Financial Capital to support PA.

### 4.1. Social Norms about PA

There were Cultural Capital differences regarding the social norm of engaging in a multitude of outdoor, lifetime PA as practiced by adults. While walking was the most common form of PA in all three counties, a broader set of activities, nearly all of which are engaged in outside (e.g., running, bicycling, swimming, and golf), was common among adults in the HPAs. Our understanding of the prevalence of specific types of PA by rurality could be enhanced by using the items that were first used in the 2011 BRFSS PA Rotating Core [46]. Further qualitative exploration to understand the types of PA that are culturally relevant and engaged in by adults in rural areas is worth undertaking because (a) the notion that “outsiders” engaged in sports outside the cultural norm was noted in the data (e.g., owners of second homes in HPA1 brought a culture of road cycling with them); (b) competitive scholastic sports such as football and basketball dominated the culture; and (c) the term “leisure” was commonly associated with sedentary time or time spent in less physically active pursuits. This highlights the importance of the degree to which the key people (Human Capital) and the influential organizations (Organizational Capital) focus on lifetime PA for adults relevant for the ages, race/ethnicities, and abilities represented in their community. The role of teachers, particularly physical educators, in helping young adults transition out of secondary education and scholastic sports to engage in lifetime PA is critical, as highlighted in seminal works about the role of physical education in public health 20–30 years ago [47,48].

Our findings regarding the social norms embedded in Cultural Capital support recent qualitative work by White and colleagues [49] using Systematic Cultural Observation methods, which demonstrated the importance of nostalgia in framing the discussion of PA in rural areas. That is, the evolution from farm-based manual labor to more sedentary occupations influences what is deemed acceptable: “the use of a nostalgic framework may mean that community members either miss or fail to welcome new, positive opportunities, such as community gardens, free gyms, and public open space—even if available” [49] (p. 131). The authors also stress the importance of assessing rural social and cultural influences of PA “through the lens of the inhabitants of rural areas” to develop culturally tailored and acceptable interventions [49]. As rural areas shift from engaging in PA as part of an occupation (e.g., farming and mining) to engaging in PA as part of “leisure-time”, embracing the dominance of hunting and fishing as a culturally appropriate leisure time activity to encourage PA could be valuable. For example, a heart attack prevention program targeting hunters, such as the Mayo Clinic’s “Is Your Heart Ready for Hunting?” [50] could be an effective, culturally appropriate program to engage rural adults to be more physically active year-round.

Competitive youth sports dominated the culture of all three counties and were supported through scholastic and other organized youth sports leagues (Organizational Capital). Despite this, only a few community sports opportunities (e.g., softball and golf) were available for adults in the HPAs, and none were available in the LPA. Studies with relatively small samples in rural states have revealed that the lack of organized sports opportunities (a) is associated with greater risk of poor cardiovascular health at a county level [51] and (b) is a barrier to rural adults utilizing existing outdoor facilities [49]. These findings suggest that utilizing a Sport for Development (SFD) model could have positive impacts on rural communities. As opposed to traditional sport programs, SFD initiatives are intentionally designed to achieve specific health goals. SFD initiatives have gained traction primarily in developing countries to address community health issues [52], but have increasingly been adopted in Western countries (e.g., Australia and Canada) to address the lack of opportunities for physical activity among marginalized populations and especially among women [53]. For rural populations in particular, SFD approaches have “demonstrated efficacy in building local skills, knowledge, and resources, increasing social cohesion, facilitating structures and mechanisms for community dialog, leadership development, and encouraging civic participation”, which could lead to increased local capacity to sustain broader sports programs [54] (p. 6). To ensure the efficacy of the SFD approach, sport should be accessible and aligned with community needs, should be adaptable and evolving rather than fixed solely in tradition, and should leverage partnerships between local and outside agencies to promote sustainability [55]. Overall, SFD could be another avenue through which rural communities could leverage their assets, and local culture could be embraced to support older youth and adult PA in the transition out of scholastic sports.

### 4.2. Quantity and Quality of Built Capital

The disparity in quantity and quality/focus of Built Capital to support the multitude of activities was also a notable difference in the HPAs. Specific differences were observed in the number of outdoor spaces for PA and the amenities at those facilities to allow for PA during evening hours and during hot, sunny, and/or rainy periods that are common in the region where the data were collected. This supports recent findings that the prevalence of walking for transportation and leisure in rural areas is similar to that of urban areas when a similar quantity of walking supports (e.g., sidewalks, paths, and trails) and destinations are present [56]. It also confirms the literature regarding the built environment factors that are frequently associated with adult PA in rural areas, including paths/trails, sidewalks, recreation facilities, schools and other publicly funded facilities, and parks [5,6,7,57,58]. The importance of the quality of facilities and amenities reiterates findings about PA in recreation areas [5] and on trails [59] in rural areas, though the amenities needed to facilitate PA during poor weather will vary based on region (e.g., rain/heat in the south and snow/cold in the north).

### 4.3. Quantity and Track Record of Human and Organizational Capital

The third critical difference in the HPAs was having more Human and Organizational Capital focused on PA, with a longer track record of PA promotion, than the LPA. Teachers, Cooperative Extension agents, and school administrators were the key people promoting PA in all three counties. The differences noted were the length of service of those key people in the HPAs (15+ years) compared with the LPA (~5 years for three of the key personnel) and how changes in Human Capital can have meaningful impacts over time on community PA. This exemplifies how important Human Capital is in small, rural towns and supports the training of trusted local individuals in PA-specific knowledge, skills, and abilities as community health workers [60]. This has been done with “promotores” in Texas-Mexico border region Latinx communities [61] and with barbers and hair stylists in Black communities across the US [62,63,64] to address a multitude of health disparities. Considering again the Mayo Clinic’s “Is Your Heart Ready for Hunting?” program [50], vendors selling hunting and fishing licenses or equipment could be trained to conduct heart disease screenings and promote PA as part of the prevention of heart attacks.

The most apparent difference in Organizational Capital was the existence of at least one municipal-level Parks and Recreation Department in each of the HPAs, but none in the LPA. The resulting difference in the quantity and quality of parks (Built Capital) was readily apparent. The HPAs also had medical and church leaders that were supportive of youth PA via programming and policies at their facilities. The practices and programming of the schools in the HPAs also differed from the LPA; schools in the HPA were more collaborative with community members in encouraging PA on school grounds. These examples demonstrate how communities can enlist more of the eight key societal sectors as recommended in the US National Physical Activity Plan (i.e., public health; healthcare; education; mass media; business and industry; not-for-profit organizations; recreation, fitness, and sports; and transportation and community planning) [65,66,67]. Constraints on workforce activities due to organizational focus on substance use disorders and social services greatly limited the focus on PA in the LPA. With the rising impact of substance use disorder in rural communities [68], the LPA that was grappling with this issue might be representative of the barriers to promoting PA in other rural counties across the US struggling with substance use disorder. Thus, a focus on PA planning and programming as adjuvant to substance use disorder treatment or prevention could hold promise [69,70]. It is also worth noting that there was not a single mention of the involvement of a local health department (LHD) in any of the counties. This could reflect (a) the lack of capacity common in rural LHDs [71], (b) a choice in organizational focus by the LHDs, which does not include the social determinants of health, and/or (c) the lack of influence of LHDs in rural areas. All topics are worth exploring.

### 4.4. A Hypothetical Causal Model of How the Forms of Community Capital Create Physically Active Rural Communities

This study sought to identify differences between LPA and HPA rural communities to determine potentially causal relationships. We consider the dependent variable (i.e., county-level PA prevalence) to be an aggregation of individual behaviors that appear to be associated with a complex set of independent variables analyzed using operationalized definitions of the eight forms of Community Capital. Taken together, these forms of Community Capital are positively related to PA, an aspect of Human Capital. This pattern fits with the notion of “spiraling up” [10] and the idea that there are many interrelated social, economic, and environmental determinants of health. From a theoretical perspective, our findings support the potential to develop a multi-level causal model using the CCF to explain the determinants of PA in rural places as described below and depicted in Figure 1. Positive attitudes toward multiple types of PA (Cultural Capital) lead to individual PA (Human Capital) and influence the use of knowledge, skills, and abilities of key leaders (Human Capital) and the policies and programs of key institutions (Organizational Capital). Human and Organizational Capital result in additional support of PA through (1) greater collaboration across organizations (Social Capital), (2) increased utilization of influential power structures to support PA (Political Capital), (3) greater investment of Financial Capital to support PA, and (4) greater quantity and quality of Built Capital to enable PA. Together, as independent variables, these forms of Community Capital determine community-level PA (i.e., the aggregation of individual community members’ PA) as a dependent variable, with reinforcing impacts on Cultural and Human Capital. This establishes PA (Human Capital) as both a determinant and result, with mutual influence, similar to the reciprocal determinism of social learning theory [72] and ecological models [73].

As an empirical example of this causal chain, the high school coach/physical education teacher in HP1, because of the culture of being active in his county, prioritized a multitude of activities (i.e., golf, watersports, running, and tennis) in his own life (Human Capital). This, in turn, led to him using his skills and abilities (Human Capital) and position in the school system in which he works (Organizational Capital) to (1) influence school district decisions (Political Capital) about sports facilities (Built Capital); (2) encourage community use of the football field/track where he coaches (Social Capital); (3) collaborate with key individuals (Social and Political Capital) to raise funds for a golf course (Financial and Built Capital); and (4) collaborate with the Cooperative Extension Agent (Organizational Capital) to implement PA programming in the school. When aggregated to a county level, these may culminate in the multidimensional support of PA, helping communities create a self-reinforcing cyclical culture of PA that is heavily reliant on Cultural Capital to expand the supply of PA-focused individuals and leaders (Human Capital) as well as groups (Organizational Capital) in rural communities.

Despite this example from our findings, multiple limitations to the potential generalizability of these findings are worth noting. First, data collection was limited to three counties in one state of the US. Despite choosing three rural counties that were diverse socioeconomically, these counties do not represent the entirety of rural America or the state. Relatedly, the use of snowball sampling led by the local Cooperative Extension Agent may have biased the findings by limiting the background and experiences of the individuals interviewed, despite the local buy-in that was afforded by such a method. Third, the primary data collector was not from the region where the data were collected, which may have limited what the interviewees were willing to share. Attempts to address this possible limitation were made by including a second data collector who was from the region and by seeking the help of the local Cooperative Extension Agent in recruiting local interviewees.

Even with these limitations, our study’s methods are worth replicating with larger samples and in socially and geographically distinct rural regions throughout the US and/or across the globe, to more confidently identify the determinants of adult PA levels in rural areas. Our results highlight areas for exploration in future research and practice in the development of active, healthy rural communities. Our findings point to a potential reinforcing feedback loop [74], or “spiraling-up” effect [10], that physically active rural communities may create. By producing greater numbers of more active and healthy people and greater amounts of aggregate PA, other forms of capital may be further strengthened, leading to additional increases in community levels of PA. Future studies of physically active rural communities (Positive Outliers) are critical to test this. Conversely, there is also the potential for a “spiraling-down” effect caused by highly inactive rural communities (Negative Outliers). Our results suggest that systems-level interventions to increase Human and Organizational Capital through rural PA coalitions may be effective in breaking that downward spiral. Thus, future evaluations of rural PA coalitions, such as *ALProHealth* in Alabama [75], *Community Coalitions for Change* in Tennessee [76], and *Heartland Moves* in Southeast Missouri [77], will be important for public health practice and rural-specific theory development. Specifically, articulating the nature of the relationships among the forms of community capital will be of particular importance, so that the causal model for creating physically active rural communities presented herein can be further refined and put into practice.

## 5. Conclusions

Rural Americans suffer from disparities in engaging in critical health behaviors, including PA, possibly in a reinforcing “spiraling-down” manner, where inactive residents and Community Capitals mutually reinforce one another in a causal model, reducing individual- and community-level PA. Further refining our understanding could be accomplished by using this causal model in future studies to develop testable hypotheses about interrelationships and reciprocal determinism of PA with the various forms of Community Capital. Specifically, future research could help us develop our understanding of how the various forms of Community Capital and PA might interact in a positive direction to counteract this downward spiral to answer questions about the ways that various forms of Community Capital positively influence PA rates and whether there are improvements or increased investments in other forms of Community Capital as community PA increases.

## Figures and Tables

**Figure 1 ijerph-18-10574-f001:**
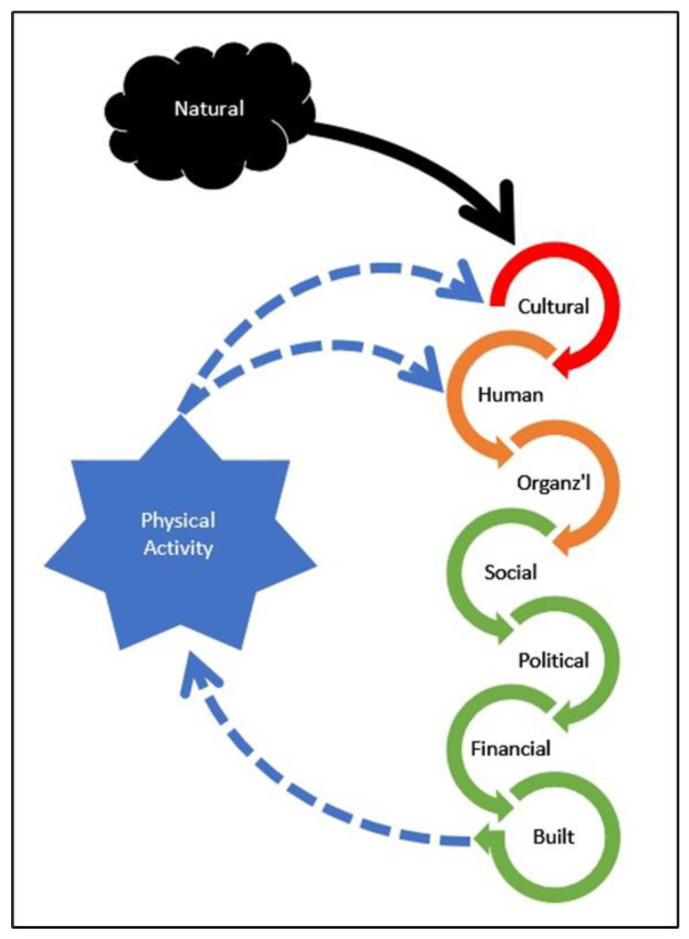
A hypothetical cyclical model of how the forms of community capital create physically active rural communities. Note. Different colors (red, orange, green) represent different levels on the causal chain.

**Table 1 ijerph-18-10574-t001:** Characteristics of the three rural southern US counties where data were collected, 2017.

Characteristic	HPA 1	HPA 2	LPA
Population ^1^	10,497	4087	8865
White, Non-Hispanic, % ^3^	78.0	42.8	68.8
Black, Non-Hispanic, % ^3^	0	0.7	21.7
Hispanic, % ^3^	18.9	55.5 ¢	6.7
Female PAG prevalence, %, 2011 [23]	56.0	49.9	39.6
Male PAG prevalence, %, 2011 [23]	59.5	56.8	44.7
Population living in a rural area, % ^1^	100	100	100
Rural-Urban Continuum Code (RUCC) ^2¶^	8	8	9
Urban Influence Code (UIC) ^2§^	4	6	10
Rural-Urban Commuting Area (RUCA; for each Census tract within the county) ^2¥^	10, 3	2	10, 10, 3
Economy “type” ^2@^	Recreation	Farming	Non-specialized
Median household income, USD ^3^	$56,573	$48,516	$29,426
Population below the poverty level, % ^3^	9.7	10.6	26.6

Notes: PAG = Physical Activity Guideline; HPA = High Physical Activity county; LPA = Low Physical Activity county; Data sources: ^1^—US Census, 2010; ^2^—US Department of Agriculture Economic Research Service, 2013; ^3^—US Census American Community Survey (ACS), 2012–2016 5-year estimates; ¢: county population includes ~1700 institutionalized people, predominantly Hispanic men; ^¶^: 8 = Completely rural or less than 2500 urban population, adjacent to a metro area; 9 = Completely rural or less than 2500 urban population, not adjacent to a metro area; ^§^: 4 = Noncore adjacent to large metro area; 6 = Noncore adjacent to small metro area and containing a town of at least 2500 residents; 10 = Noncore adjacent to micro area and not containing a town of at least 2500 residents; ^¥^: 2 = Metropolitan area high commuting: primary flow 30% or more to an urbanized area; 3 = Metropolitan area low commuting: primary flow 10% to 30% to an urbanized area; 10 = Rural areas: primary flow to a tract outside an urbanized area or urban cluster; ^@^ Farm-dependent county indicator: Farming accounts for at 25% or more of the county′s earnings or 16% or more of the employment averaged over 2010–2012; Recreation-dependent county indicator: defined based on an index that reflects earnings and employment in selected recreation-related industries together with the percentage of vacant housing units intended for seasonal or occasional use; Nonspecialized indicator: The county was not a farming, mining, manufacturing, government-dependent, or recreation county.

**Table 2 ijerph-18-10574-t002:** Key stakeholder interviewee characteristics.

Interviewee Code	Sex	Age Group	Race	Title	Interview Length
HPA1.1	Female	40–49	White	Cooperative Extension County Coordinator	44:09
HPA1.2	Male	50–59	White	Cooperative Extension Agent	5:10
HPA1.3	Male	40–49	White	High School Coach/Physical Education Teacher	32:03
HPA1.4	Female	50–59	White	Cooperative Extension Administrative Assistant	2:40
HPA2.1	Female	40–49	White	Cooperative Extension County Coordinator	13:10
HPA2.2	Female	40–49	White	High School Principal/Superintendent	14:10
HPA2.3	Male	70+	White	Newspaper Publisher	18:40
LPA.1	Female	50–59	White	Cooperative Extension County Coordinator	15:24
LPA.2	Female	60–69	White	County Judge	42:28
LPA.3	Female	50–59	White	High School Principal	26:27
LPA.4	Female	40–49	White	School District, Counselor	31:19
LPA.5	Female	70+	White	Extension Programming Volunteer with Seniors	26:03
LPA.6	Female	40–49	White	Chamber of Commerce Staff Member	11:56
LPA.7	Male	50–59	Black	High School Teacher/Boys and Girls Club Director	30:21

**Table 3 ijerph-18-10574-t003:** Forms of Community Capital, original definitions [10] (pp. 20–21) and operational definitions utilized for the qualitative data analysis.

Form of Capital	Original Definition	Operational Definition
Cultural Capital	Cultural capital reflects the way people “know the world” and how they act within it, as well as their traditions and language. Cultural capital influences what voices are heard and listened to, which voices have influence in what areas, and how creativity, innovation, and influence emerge and are nurtured. Hegemony privileges the cultural capital of dominant groups.	The way people “know the world” that hinders or fosters how they act within it (i.e., cultural beliefs and traditions influence individual decisions about engaging in PA), as well as their language about and attitudes toward PA. Cultural capital influences whose voices are heard and which voices have influence in what areas, and how creativity, innovation, and influence emerge and are nurtured. Hegemony privileges the cultural capital of dominant groups.
Human Capital	Human capital is understood to include the skills and abilities of people to develop and enhance their resources and to access outside resources and bodies of knowledge in order to increase their understanding, to identify promising practices, and to access data for community-building. Human capital addresses the leadership’s ability to “lead across differences”, to focus on assets, to be inclusive and participatory, and to act proactively in shaping the future of the community or group.	The skills and physical abilities of people to develop and access outside resources and bodies of knowledge about PA in order to increase their understanding and identify promising practices. Human capital addresses the leadership’s ability to “lead across differences”, to focus on assets, to be inclusive and participatory, and to act proactively in shaping the future of the PA of the community or group that they have influence over. It also includes the facilitators of and barriers to using skills and abilities to affect community level PA (e.g., time).
Social Capital	Social capital reflects the connections among people and organizations or the social “glue” to make things, positive or negative, happen. Bonding social capital refers to those close redundant ties that build community cohesion. Bridging social capital involves loose ties that bridge among organizations and communities.	The connections among people and organizations or the social “glue” to make things happen that increase community PA level. Bonding social capital refers to close ties that build community cohesion. Bridging social capital involves loose ties that bridge across social groups, organizations, and communities.
Organizational Capital [36] (p. 113)	The structure, policies, plans, and track record of existing groups (informal groups, organizations, and networks).	The structure, policies, plans, and track record of existing groups (informal groups, organizations, and networks) and their ability to collaborate in supporting PA.
Political Capital	Political capital reflects access to power, organizations, resources, and power brokers. Political capital also refers to the ability of people to find their own voice and to engage in actions that contribute to the well-being of their community.	Community political power, influence, and access to power brokers at local, county, state, and federal levels who support PA.
Financial Capital	Financial capital refers to the financial resources available to invest in community capacity-building, to underwrite the development of businesses, to support civic and social entrepreneurship, and to accumulate wealth for future community development.	Financial resources available to invest in programs and infrastructure that support PA.
Natural Capital	Natural capital refers to those assets that abide in a particular location, including weather, geographic isolation, natural resources, amenities, and natural beauty. Natural capital shapes the cultural capital connected to place.	Those assets that exist in a particular location without human intervention (i.e., not parks) that either foster or hinder community level PA, including weather, topography, natural resources, and natural beauty. Natural capital influences the cultural capital connected to place.
Built Capital	Built capital includes the infrastructure supporting these activities.	Built capital includes the infrastructure supporting PA activities, including the accessibility to and presence of parks. A sense of personal safety is necessary for its use.

Notes: PA = physical activity; sources cited by Emery and Flora in the original definitions of the forms of capital were removed.

## Data Availability

Data (transcripts) are not available for public to retain confidentiality of participants and their communities.

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
