# Peer review of "What Sets Physically Active Rural Communities Apart from Less Active Ones? A Comparative Case Study of Three US Counties"

_ijerph, 2021, doi:10.3390/ijerph182010574_

Round 1

Reviewer 1 Report

Very interesting theme on the influence of different factors on physical activeness of rural population. The conclusion that by producing greater numbers of more active and healthy people and greater amounts of physical activity, other forms of capital may be further strengthened, is clear and predictable. As a tip, one can suggest adding methods of quantitative assessment to descriptive methods and transcription of interviews.

Author Response

Resubmission letter attached. Thank you.

Reviewer 2 Report

Abildso and colleagues conducted a series of stakeholder interviews, on-site intercept interviews, and in-person observations in a comparative case study in the United States to explore factors contributing to physically active communities. Overall the study findings are informative, interesting and generalisable, and are of high relevance to improving population health in rural settings.

Minor comments –

2.2 Instruments and Data Collection: the authors captured barriers that “keep people from being more active”. Were any enablers supporting and facilitating active living were captured as well? 
2.3 Analysis: please indicate if any software, such as NVivo, was used for the qualitative data analysis.
The authors have presented characteristics of key stakeholders in Table 2. However, the information of subjects participating in the in-person observations were largely absent. A little bit more details of how the in-person observations were performed would help international readers better understand the study design and the context within which the results were generated.

Suggested references to be cited –

As mentioned by the authors in the Introduction / Discussion section, a number of systems-level structural capacity determinants such as financial, organisational, and human resources could contribute to the rural-urban disparities in physical activities and the subsequent quality of life. This also applies to other developing countries, where literature suggests that factors on family and social support may also play a role. This may also echo the causal chain shown in Figure 1, and can be added to the discussions on relationship with literature. For example, an earlier study conducted in southern China found that a higher level of family and social support could contribute to enhancing physical activities and activities of daily living among rural population with long-term conditions. Suggested citation [X]: Int J Environ Res Public Health. 2015; 12(10):13209-23. doi: 10.3390/ijerph121013209.

Author Response

(The authors gave the same response as above.)

Reviewer 3 Report

Please, see the attached document.

Author Response

(The authors gave the same response as above.)

Round 2

Reviewer 3 Report

Apart from some changes in the section on conclusions, the observations made in the first review have not been addressed.

Author Response

Thank you for the additional review. We know this is a length manuscript that has take a lot of time to review and comment about.

1) May we ask for clarity with regard to the comment about the third person plural usage to help us address it: "In this editorial review exercise, the use of the third person plural must be corrected (some examples: lines 90 and 468)." We didn't see the specific instance, but will be happy to correct any concerns in the revised document being submitted

2) They should review the titles and sources of all figures and tables. The best example of this is the figure included in line 707 where both are missing.

  • Response: Apologies for this formatting issue. We have improved the formatting of Figure 1 so that the title precedes the figure and the note immediately follows it without any page breaks. Thank you.

3) Despite the length of the article, the conclusions are very scarce; perhaps as a consequence of the length of the discussion heading. The drafting of these conclusions must be reoriented and matured; It is not correct to include citations to other authors in this section. The question that ends this section is not understood. Finally, they should include a paragraph that refers to the possible limitations of their study

  • Limitations are listed in a paragraph running from line 729 to 739 in the Discussion section.
  • In previous versions that my co-authors and I have worked on we had the section "4.4. A hypothetical causal model of how the forms of community capital create physically active rural communities" as the key conclusion but it became a very length section. We chose to put that in the Discussion section because we were operating under the Instructions for Authors from IJERPH, pasted below:
    • Discussion: Authors should discuss the results and how they can be interpreted in perspective of previous studies and of the working hypotheses. The findings and their implications should be discussed in the broadest context possible and limitations of the work highlighted. Future research directions may also be mentioned. This section may be combined with Results.
    • Conclusions: This section is mandatory, and should provide readers with a brief summary of the main conclusions.
  • We are happy to move that section to the Conclusions if it is the pleasure of the editor and reviewer. 

Many thanks!